# lncRNA ch-MYC-AS1 restricts ALV-J replication by disrupting the ANXA2-C-Myc oncogenic axis

Suyu Fan,[1,2] Weiyi Zhou,[3] Xuming Hu,[1,2] Yingjie Gu,[1,2] Guangzhong Peng,[1] Yu Zhang,[1] Wenming Zhao,[1] Guohong Chen,[1,2,4] Qi Xu[1,2,4]

**ABSTRACT** The c-Myc oncogene is a critical regulator of viral oncogenesis and immune evasion in multiple cancers. However, its modulation by long noncoding RNAs (lncRNAs) during retroviral infection remains poorly understood. Here, we show that ch-MYC-AS1, a novel lncRNA, restricts avian leukosis virus subgroup J (ALV-J) replication by targeting c-Myc protein expression using chicken macrophage HD11 cells. Mechanistically, ch-MYC-AS1 binds to annexin A2 (ANXA2) and impedes its nuclear translocation, preventing its collaboration with c-Myc to promote glycolysis. This dual inhibition suppresses c-Myc-driven metabolic reprogramming essential for viral proliferation. Our findings reveal ch-MYC-AS1 as a key suppressor of retroviral replication through coordinated disruption of c-Myc/ANXA2 signaling, providing a potential therapeutic strategy for antiviral and anticancer drug development.

**IMPORTANCE** Avian leukosis virus subgroup J (ALV-J) is an oncogenic retrovirus that causes tumors and immunosuppression in chickens, leading to significant economic losses in poultry industries. This study identifies a host-derived long noncoding RNA (lncRNA), ch-MYC-AS1, which suppresses ALV-J replication by disrupting the ANXA2-c-Myc signaling axis. These findings unveil a novel layer of antiviral defense mediated by an epigenetically regulated lncRNA and highlight a potential RNA-based strategy to combat retroviral infections. Moreover, targeting the ANXA2-c-Myc interaction may offer therapeutic insights for controlling ALV-J and other MYC-driven diseases.

**KEYWORDS** oncogenic retrovirus, ch-MYC-AS1, ANXA2, avian leukosis virus

Oncogenic retroviruses, originally discovered in chickens, have provided fundamental models for cancer biology, including the discovery of viral oncogenes and their cellular counterparts, the proto-oncogenes (1). Among these, research on avian leukosis virus (ALV)-induced tumors led to the identification of the cellular proto-oncogene c-myc (2). Subsequent work established c-myc as a common retroviral integration site whose activation drives tumorigenesis across diverse species, including in models mediated by ALV (3, 4), feline leukemia virus (5), murine leukemia retrovirus (6), and human T-cell leukemia retrovirus (7). The c-Myc oncoprotein is a master regulator of numerous cellular processes, including cell proliferation, metabolism, and, more recently, immune modulation. It has been shown to orchestrate tumor immune evasion by regulating key immune checkpoints such as CD47 and PD-L1 (8), highlighting its potential as a therapeutic target in oncology (2, 8–10).

Beyond its well-characterized genetic regulation, emerging evidence implicates the importance of epigenetic mechanisms in c-myc regulation. Bidirectional transcription at the c-myc locus is evolutionarily conserved in humans, mice, rodents, and cattle (11–14), producing both sense and antisense transcripts. Notably, natural antisense transcripts originating from the 3′ distal region of the c-myc locus have been implicated in a sophisticated regulatory cascade fine-tuning c-myc levels (15). Furthermore, treatment

**Peer Reviewer** Wentao Li, Huazhong Agricultural University, Wuhan, Hubei, China

Address correspondence to Xuming Hu, hxm@yzu.edu.cn, or Qi Xu, xuqi@yzu.edu.cn.

The authors declare no conflict of interest.

See the funding table on p. 14.

with DNA methyltransferase inhibitors such as the DNA methylation inhibitors 5-azacytidine (5-Aza-CR) and 5-aza-2′-deoxycytidine (5-Aza-CdR) can suppress c-Myc protein expression, implicating DNA methylation in its epigenetic control (16). These findings suggest that epigenetic silencing of regulatory antisense RNAs may be a mechanism for c-myc activation during tumorigenesis. In chickens, several long non-coding RNAs (lncRNAs) serve as key regulators of antiviral immunity by modulating interferon responses and viral replication pathways (17–19). However, whether antisense noncoding RNAs from the chicken c-myc locus possess intrinsic antiviral functions against retroviruses that dysregulate this oncogene remains unexplored.

In this study, we aimed to characterize antisense RNAs transcribed from the chicken c-myc proto-oncogene and investigate their role in the host defense against the oncogenic retrovirus ALV subgroup J (ALV-J). We identified a novel long noncoding RNA (lncRNA), termed ch-MYC-AS1, and demonstrated that it functions as a key suppressor of ALV-J replication by disrupting the ANXA2-c-Myc oncogenic axis. Notably, Annexin A2 (ANXA2) serves as a key cellular receptor for ALV-J entry (20) and also functions as an RNA-binding protein implicated in post-transcriptional regulation, including that of c-Myc (21), highlighting its potential central role in virus-host interactions. Our findings reveal a novel layer of host antiviral defense mediated by an epigenetically regulated lncRNA, providing new insights into the intricate interplay between retroviruses and host regulatory networks.

## MATERIALS AND METHODS

### Cells, viruses, and plasmids

The chicken macrophage cell line HD11 cells were derived from chicken bone marrow and transformed with the avian myelocytomatosis virus MC29 (22) and maintained in Dulbecco's Modified Eagle Medium (DMEM)-high glucose or DMEM-low glucose (Gibco, USA) with 10% fetal bovine serum (FBS) at 41°C, 5% $CO_2$, and 95% humidity. The virus strain JS09GY3 (GenBank accession number GU982308) of avian leukosis virus subgroup J (ALV-J) was isolated from field-infected commercial layer chickens with hemangioma, myeloid leukosis, and an insertion in the E element (23). The plasmids pcDNA3.1-ch-MYC, pcDNA3.1-ch-MYC-AS1, pcDNA3.1-ch-ANXA2, and pcDNA3.1-EGFP came from the plasmid bank in our laboratory.

### 5-Aza-dC treatments

HD11 cells were seeded into six-well plates and treated for 48 h with 1 µM 5-aza-2′-deoxycytidine (5-Aza-Dc; cat#A3656, Sigma-Aldrich), an inhibitor of DNA methyltransferases. The 5-Aza-dC dose was selected based on previous study and our preliminary data, and DMSO treatment was used as a control.

### Plasmids transfection

HD11 cells were transfected with the pcDNA3.1-EGFP or pcDNA3.1-ch-MYC-AS1 or pcDNA3.1-ch-MYC plasmid using Lipofectamine 3000 Transfection Reagent (Thermo Fisher Scientific, USA) for 36 h, and then total RNA and protein were collected for gene expression analysis.

For viral infection experiments, HD11 cells were first infected with the ALV-J virus at a multiplicity of infection (MOI) of 0.2, 1, and 5 for 12 h to allow viral adsorption and entry. Cells were then transfected with pcDNA3.1-EGFP or pcDNA3.1-ch-MYC-AS1 or pcDNA3.1-ch-MYC or pcDNA3.1-ch-ANXA2 plasmid using Lipofectamine 3000 Transfection Reagent (Thermo Fisher Scientific, USA) for another 36 h and then collected for ALV-J proliferation analysis.

## RNA interference assays

Gene-specific antisense oligonucleotides (ASOs) targeting ch-MYC-AS1 and small interfering RNAs (siRNAs) targeting chicken MYC were designed and synthesized by Guangzhou RiboBio Co., Ltd. (Guangzhou, China). The sequences were as follows:

ch-MYC-AS1 ASOs: ASO-1 (5′-GGACUCUGUGCUGGAUUCAG-3′),

ASO-2 (5′-CCUUCUCGUUGUUGGCCACC-3′),

ASO-3 (5′-UCCUCUGAGUCUAACGUGCG-3′).

Chicken MYC siRNAs: siRNA-1 (5′-CCAGCAAGAACUACGAUUA-3′),

siRNA-2 (5′-CAGACUAAUCGCAGAGAAA-3′), and

siRNA-3 (5′-AGAUCAGCAACAACCGAAA-3′).

For the knockdown of ch-MYC-AS1 or chicken MYC, HD11 cells were transfected with ch-MYC-AS1 ASOs, chicken MYC-specific siRNAs, or the control using Lipofectamine RNAiMAX Transfection Reagent (Thermo Fisher Scientific, USA) for 36 h. Total RNA and protein were collected for gene expression analysis.

For viral infection experiments, HD11 cells were first transfected with ch-MYC-AS1 ASOs, chicken MYC siRNAs, or the control using Lipofectamine RNAiMAX Transfection Reagent (Thermo Fisher Scientific, USA) for 12 h to allow for adequate expression prior to ALV-J infection. Cells were then infected with the ALV-J virus at an MOI of 5 for another 48 h and then collected for ALV-J proliferation analysis.

## Rapid amplification of cDNA ends

The 5′ and 3′ rapid amplification of cDNA ends (RACE) experiments were performed using the SMARTer RACE 5′/3′ Kit (Takara, Clontech Laboratories, 634,860) following the manufacturer's instructions. Briefly, total RNA (the genomic DNA was removed) was converted into RACE-Ready first-strand cDNA using the 5′ or 3′ CDS Primer A (provided by the kit). Next, 5′ RACE PCR amplification was conducted with the 5′ gene-specific primers TCAGAGGAGAACGACAAGAGGCGAACG and universal primer (provided by the kit) to generate the 5′ cDNA fragments. The 3′ RACE PCR amplification was conducted by the 3′ gene-specific primer: TCCTCCGCCTCAACTGCTCTTTCTCTG and universal primer. The 5′ and 3′ RACE products were further characterized by 1.5% agarose gel electrophoresis and sequenced by TA cloning. After reaching the 5′ and 3′ cDNA ends, the full-length ch-MYC-AS1 transcripts were amplified using the following primers: forward 5′-GATTCTAAGTGATGTCCAAG-3′ and reverse 5′-TTTTTCTTCCGACACGCC-3′ and subsequently cloned into the TA cloning vector for sequencing.

## Northern blot

Northern blotting assays were performed with the digoxigenin (DIG) Northern Starter Kit (cat#12039672910, Roche) according to the manufacturer's instructions. In brief, the full-length DNA template of ch-MYC-AS1 was first prepared by PCR amplification with the following primers: forward 5′-GTATTCTAAGTGATGTCCAAG-3′ and reverse 5′-<u>TAATA CGACTCACTATAGGG</u>AGAACAAGAAGAAGATGAGG-3′. A digoxigenin (DIG)-labeled RNA probe was then prepared using *in vitro* transcription with the T7 polymerase. Next, total RNA was separated by 2% formaldehyde gel electrophoresis at 50 V for 6 h and then transferred to positively charged nylon membranes (11209299001, Roche) by capillary transfer with 20× saline sodium citrate (SSC) overnight. After RNA fixation by baking at 80℃ for 2 h, blots were hybridized with denatured DIG-labeled RNA probe overnight at 68℃. Finally, blots were incubated with anti-digoxigenin-AP antibody for 30 min at room temperature and developed using the ready-to-use chemiluminescence substrate CDP-Star on the FluorChem Q imaging system (Protein Simple).

## Coding potential analysis of the antisense RNA ch-MYC-AS1

Full-length ch-MYC-AS1 was cloned into the eukaryotic expression vector pcDNA3.1 with the N-terminal start codons ATG and HA tag in all three coding possibilities, and the plasmids were subsequently transfected into DF-1 cells. The primers used can be found

in Table 1. P53 was cloned into the pcDNA3.1 vector and used as a positive control. After 48 h, immunoblotting was used to detect the HA tag.

## RNA fluorescence *in situ* hybridization

RNA fluorescence *in situ* hybridization (FISH) was performed according to the manufacturer's recommendations (RIBOBIO, Guang Zhou). We routinely ordered sets of fluorescent FISH probe mixes for ch-MYC-AS1 from commercial sources (RIBOBIO). To achieve a sufficient signal-to-background ratio, multiple probes were targeted along each individual lncRNA sequence. A set of 15–20 probes that cover the entire length of the RNA molecule provided an optimal signal strength, and each probe carried multiple fluorophores. The pooled FISH probes were resuspended to a final concentration of 25 µM in RNase-free storage buffer and protected from light at −20°C. Cells were fixed with 4% paraformaldehyde in phosphate-buffered saline (PBS) for 20 min at room temperature, permeabilized with 0.25% Triton X-100 for 5 min, and blocked with 2% BSA for 30 min. Cells were then incubated overnight with the ch-MYC-AS1 probe mix at 37°C. For colocalization studies, after RNA-FISH, cells were subjected to immunofluorescence. The pictures were captured with a Leica SP8 confocal microscope (X100) and merged.

## RNA pull-down

RNA pull-down experiments were performed using the Pierce Magnetic RNA-Protein Pull-Down Kit (20164, Thermo Scientific). First, the full-length DNA template of ch-MYC-AS1 was prepared from HD11 cells by PCR amplification with the following primers: forward 5′-GTATTCTAAGTGATGTCCAAG-3′ and reverse 5′-<u>TAATACGACTCACTATAGGG</u>TTT TTCTTCCGACACGCC-3′. The antisense template of ch-MYC-AS1 was prepared by PCR amplification with the forward primer 5′-<u>TAATACGACTCACTATAGGG</u>GTATTCTAAGTGATGT CCAAG-3′ and reverse primer 5′-TTTTTCTTCCGACACGCC-3′. Next, RNA synthesis *in vitro* transcription was performed using the T7 High Yield RNA Synthesis Kit (New England Biolabs, E2040S), and RNA purification was performed by a Monarch RNA Cleanup Kit (New England Biolabs, T2040). The 3′ terminus of an RNA strand was then biotinylated by the Pierce RNA 3' End Desthiobiotinylation Kit (20163, Thermo Scientific). The whole-cell lysate from HD11 cells was harvested and resuspended in Pierce IP Lysis Buffer (87787, Thermo Scientific) containing RNase and protease inhibitors. Biotinylated RNA (50 pmol)

**TABLE 1** Primers used in this study[a]

| Primer name | Primer sequences |
|---|---|
| ch-MYC-AS1 fwd | CACAGACTAATCGCAGAGA |
| ch-MYC-AS1 rev | GTGATGTCCAAGAGTTCCTA |
| ch-MYC-AS1 RT primer | CAGAGGAGAACGACAAGAGGCGAAC |
| ch-GAPDH RT primer | CCAAACTCATTGTCATACCAGGAAAC |
| ch-β-actin RT primer | CGTACTCCTGCTTGCTGATCCACAT |
| ALVJ env fwd | TTGGTTCGGTGTGCTATG |
| ALVJ env rev | GTCTCGTTGCTGGTGAAT |
| ch-β-actin fwd | GAGAAATTGTGCGTGACATCA |
| ch-β-actin rev | CCTGAACCTCTCATTGCCA |
| ch-GAPDH fwd | GAGAAACCAGCCAAGTATGA |
| ch-GAPDH rev | CTGGTCCTCTGTGTATCCTA |
| HA-ch-MYC-AS1-F | CCC<u>aagctt</u>atggcctacccctacgacgtgcccgactacgcc GTATTCTAAGTGATGTCCAAG |
| HA-T-ch-MYC-AS1-F | CCC<u>aagctt</u>atggcctacccctacgacgtgcccgactacgcc TGTATTCTAAGTGATGTCCAAG |
| HA-TT-ch-MYC-AS1-F | CCC<u>aagctt</u>atggcctacccctacgacgtgcccgactacgcc TTGTATTCTAAGTGATGTCCAAG |
| HA-ch-MYC-AS-R | TGC<u>tctaga</u>TTTTTCTTCCGACACGCC |

[a]The underlined sequences indicate the recognition sites for the restriction endonucleases HindIII (AAGCTT) and XbaI (TCTAGA).

was then added to 50 µL of prewashed Pierce Nucleic-Acid Compatible Streptavidin Magnetic Beads and incubated for 30 min at room temperature with agitation. The beads were mixed with 200 µg of the whole-cell lysate and incubated for 60 min at 4°C with rotation. The beads were washed three times with wash buffer and then incubated for 30 min at 37°C with elution buffer. The eluted protein samples were separated on a 12% SDS-PAGE gel and stained with a Fast Silver Stain Kit (P0017S, Beyotime) according to the manufacturer's instructions. Protein in gel slices was digested with trypsin and identified using nano-high-performance liquid chromatography mass spectrometry (MS) in the Proteome Research Center of Fudan University.

## RNA-binding protein immunoprecipitation

Whole-cell lysate from HD11 cells was harvested and resuspended in Pierce IP Lysis Buffer (87787, Thermo Scientific) containing RNase and protease inhibitors. A total of 10 mg protein lysate was combined with 10 µg rabbit polyclonal to Annexin-2/ANXA2 antibody (ab235939, Abcam) or rabbit IgG (ab172730, Abcam) in 500 µL ice-cold RNA-binding protein immunoprecipitation (RIP) buffer and incubated overnight at 4°C with rotation. Next, the protein sample/antibody mixture was added to a 1.5 mL microcentrifuge tube containing 50 µL prewashed Pierce Protein A/G Magnetic Beads and incubated for 1 h at 4°C with gentle rotation. After washing, the beads were resuspended in 1 mL TRIzol RNA extraction reagent, and then the coprecipitated RNA was isolated according to the manufacturer's instructions. Finally, the expression of antisense RNA ch-MYC-AS1 in the precipitates was detected by strand-specific RT-qPCR.

## Enzyme-linked immunosorbent assay

HD11 cell culture supernatants were collected, and the p27 antigen of ALV-J in the cell supernatants was measured by enzyme-linked immunosorbent assay (ELISA; IDEXX, Beijing, China) following the manufacturer's instructions.

## TCID$_{50}$ assay

DF-1 cells were seeded in growth media on each well of 96-well plates and cultured in DMEM with 5% FBS at 37°C in 5% $CO_2$ and 95% humidity. After preparing the series of dilutions at 1:10 of the original virus sample, mildly add 0.1 mL of virus dilution per well, infecting 8 wells per dilution, and then culture for 1 week at 37°C. The viral replication in the DF1 cells was analyzed by immunofluorescence assay (IFA) using mouse monoclonal antibody JE9, which is specific to the envelope protein of ALV-J at 1 week post-inoculation. The viral titer is calculated using the method of Muench and Reed.

## Reverse transcription-quantitative PCR

Reverse transcription-quantitative PCR (RT–qPCR) assays were performed according to previous studies (24). Briefly, total RNA was extracted from chicken cells or tissues using TRIzol reagent (Thermo Fisher Scientific, USA) according to the manufacturer's recommendations. The gDNA Eraser-treated RNA samples were reverse transcribed with RT primers at 37°C for 15 min or strand-specific RT primers at 42°C for 15 min with PrimeScript Reverse Transcriptase (TaKaRa, Japan). Quantitative PCR was then performed with gene-specific primers and SYBR Green Master Mix (TaKaRa, Japan) on the CFX Connect Real-Time PCR Detection System (Bio–Rad, California, USA). GAPDH and β-actin RNA levels were used as internal controls to normalize gene expression. The gene-specific primers are listed in Table 1.

## Western blotting

Protein was separated by 10% SDS-PAGE at 120 V for 90 min and then transferred to polyvinylidene difluoride membranes at 50 V for 150 min. Membranes were blocked in TBS-T containing 5% nonfat dry milk (BIO-RAD). Primary antibodies were incubated

overnight at 4°C with agitation. The following antibodies were used to determine protein expression: rabbit polyclonal to beta Tubulin (ab6046, Abcam), rabbit polyclonal to Lamin B1 (ab16048, Abcam), rabbit monoclonal to GAPDH (ab181602, Abcam), rabbit polyclonal to Annexin-2/ANXA2 antibody (ab235939, Abcam), mouse monoclonal to c-Myc (ab56, Abcam), and mouse monoclonal antibody JE9, which is specific to the envelope protein of ALV-J. After washing extensively with TBST, secondary antibodies (anti-rabbit or anti-mouse horseradish peroxidase conjugate, 1:10,000 dilution) were incubated for 1 h at room temperature. After washing extensively with TBST, blots were developed using enhanced chemiluminescent detection reagents on the FluorChem Q imaging system (Protein Simple).

## Co-immunoprecipitation

Co-immunoprecipitation (Co-IP) was performed using the Thermo Scientific Pierce Classic Magnetic IP/Co-IP Kit to confirm the interaction between c-Myc and ANXA2 protein. Briefly, both input and IP samples were analyzed by western blotting using the rabbit polyclonal to Annexin-2/ANXA2 antibody (ab235939, Abcam) and mouse monoclonal to c-Myc (ab56, Abcam).

## Cytoplasmic and nuclear protein extraction

Cytoplasmic proteins and nuclear proteins were extracted from different chicken cells using the NE-PER Nuclear and Cytoplasmic Extraction Reagents Kit (78833, Thermo Scientific). The extracted proteins were then confirmed by western blotting for beta Tubulin and Lamin B1 expressed in the cytoplasm and in the nuclei.

## Statistical analyses

The statistical analysis was performed with the Statistical Package for the Social Sciences (version 16.0) software. Statistical significance was assessed using a two-tailed unpaired Student's $t$-test with a $P$ value threshold of <0.05.

## RESULTS

### Identification of proto-oncogene c-myc-derived antisense RNAs

To identify the proto-oncogene c-myc-derived antisense RNAs, a double-stranded RNA library was constructed using macrophages, which were treated with or without the DNA methylation inhibitor 5-Aza-CdR, and further analyzed the sequences by high-through-put next-generation sequencing (Fig. 1A). In this study, we identified one antisense RNA derived from exon 3 region of chicken c-myc gene and named ch-MYC-AS1 (Fig. 1B). By 5′ and 3′ rapid amplification of cDNA ends (RACE) analysis, we first identified the ch-MYC-AS1 is 621 bp in length, which was further validated by northern blot analyses (Fig. 1C through E). Additionally, we confirmed that ch-MYC-AS1 has no coding capacity by *in vitro* translation analysis (Fig. 1F), indicating that it is a long noncoding RNA (lncRNA).

### Overexpression of ch-MYC-AS1 inhibits ALV-J proliferation

To investigate the role of ch-MYC-AS1 in antiviral immunity, we first transfected the chicken macrophage cell line HD11 with a lentivirus-mediated expression system containing the ch-MYC-AS1 and then infected oncogenic retrovirus ALV-J for 48 h. It was found that overexpression of ch-MYC-AS1 in HD11 cells significantly inhibited the expression of ALV-J *env* gene at both mRNA and protein levels at a multiplicity of infection (MOI) of 0.2, 1, and 5 (Fig. 2A and B). $TCID_{50}$ and ELISA analysis results also showed that ch-MYC-AS1 led to a significant reduction of viral titers and viral protein in the culture medium of macrophages infected with ALV-J at an MOI of 5 (Fig. 2C and D). The confocal immunofluorescence microscopy analysis further confirmed that inhibition

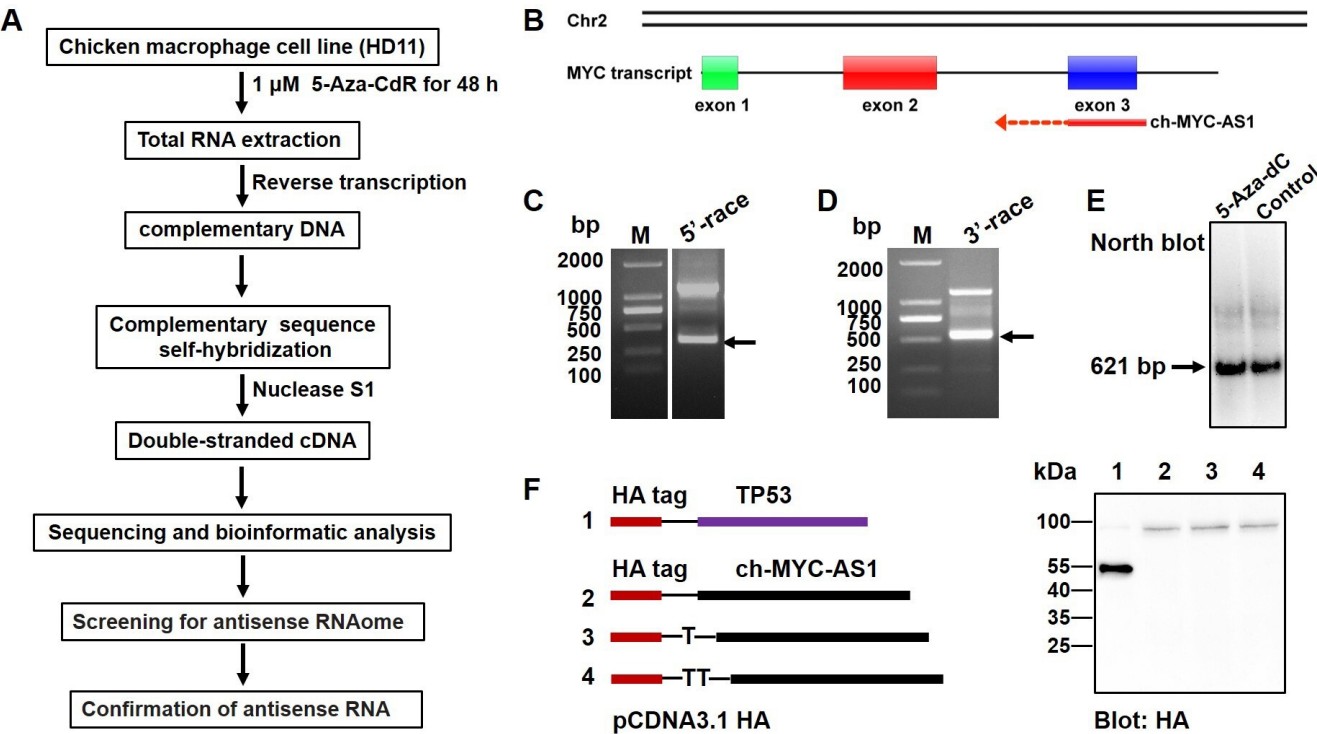

FIG 1 Identification of ch-MYC-AS1, an antisense RNA of chicken proto-oncogene c-myc. (A) Schematic diagram of screening for antisense RNAome protocols used. (B) Schematic overview of ch-MYC-AS1 from the proto-oncogene c-myc in the chicken genome. (C) Transcriptional start site and (D) end site were separately identified with a 5′ cap adapter and a 3′ poly(A) adapter using special primers in a RACE assay that was performed with HD11 cells treated with 5-Aza-dC. (E) Northern blotting analysis of ch-MYC-AS1 expression in HD11 cells treated with 5-Aza-dC. (F) Coding potential analysis of ch-MYC-AS1. Full-length ch-MYC-AS1 was cloned into the eukaryotic expression vector pcDNA3.1 with an N-terminal start codon (ATG) and an HA-tag in all three reading frames, and the plasmids were subsequently transfected into HEK293 cells. Immunoblotting was used to detect the HA tag after 48 h. Cells transfected with a plasmid containing TP53 with an HA tag were used as a positive control.

of ALV-J proliferation was inhibited by ch-MYC-AS1 in the chicken HD11 cells infected with ALV-J at an MOI of 5 (Fig. 2E).

## Knockdown of ch-MYC-AS1 promotes ALV-J proliferation

Conversely, knockdown of ch-MYC-AS1 significantly increased the expression of ALV-J *env* gene at both mRNA and protein levels in HD11 cells at an MOI of 5 (Fig. 3A and B). TCID50 and ELISA analysis results also showed that knockdown of ch-MYC-AS1 led to a significant increase of viral titers and viral protein in the culture medium of HD11 cells infected ALV-J at an MOI of 5 (Fig. 3C and D). The confocal immunofluorescence microscopy analysis further confirmed that knockdown of ch-MYC-AS1 can promote the replication of ALV-J in macrophage (Fig. 3E). Our data collectively indicate that ch-MYC-AS1 may possess a function in antiviral defense against the oncogenic retrovirus ALV-J infection in chicken macrophages.

## ch-MYC-AS1 modulates ALV-J proliferation through c-Myc protein

Next, we evaluated whether the ch-MYC-AS1 affects ALV-J proliferation through regulating the expression of c-Myc protein. We found that the expression of c-Myc protein was obviously reduced in the highly expressed ch-MYC-AS1 chicken HD11 cells (Fig. 4A). Correspondingly, the reduction of c-Myc protein decreased the expression of ALV-J Env protein (Fig. 4A). On the contrary, knockdown of ch-MYC-AS1 by RNAi significantly increased the expression of c-Myc and ALV-J Env protein in the chicken HD11 and DF-1 cells (Fig. 4B). Consistent with these findings, the expression of ALV-J Env protein was

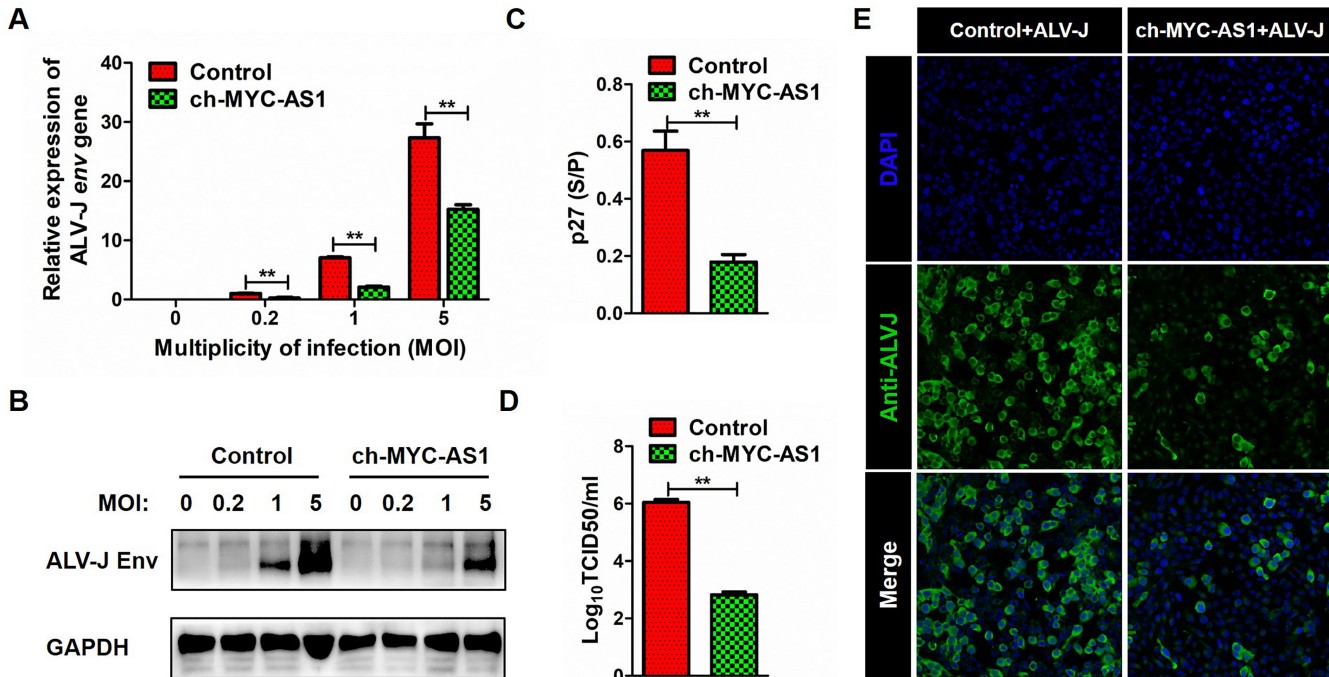

**FIG 2** ch-MYC-AS1 inhibited ALV-J proliferation in chicken macrophages. Chicken macrophage cell line HD11 cells were first transfected with a lentivirus-mediated expression system containing the ch-MYC-AS1 or control and then infected with the ALV-J at MOI of 0.2, 1, and 5 for 48 h. RT-qPCR (A), western blotting (B), and TCID$_{50}$ (C) analysis of ALV-J *env* gene expression in HD11 cells. (D) ELISA analysis of ALV-J p27 protein expression in the supernatant of HD11 cells. (E) Confocal immunofluorescence microscopy analysis of ALV-J *env* gene expression in HD11 cells. HD11 cells were incubated with the anti-ALV-J envelope protein (JE9 antibody) and then stained with goat anti-mouse IgG conjugated with the Alexa Fluor 488 dye (Sigma-Aldrich). The nuclei were stained with DAPI dye (Sigma-Aldrich). The pictures were captured and merged with a Leica SP8 confocal microscope (20×). Error bars represent the SEM, *n* = 3. *$P < 0.05$ and **$P < 0.01$ (two-tailed Student's *t*-test).

obviously decreased by knockdown of c-Myc and was promoted by overexpression of c-Myc protein in HD11 cells (Fig. 4C and D). These findings indicated that ch-MYC-AS1 could block ALV-J proliferation through modulating c-Myc protein expression.

## ch-MYC-AS1 can interact with Annexin A2

To dissect the mechanistic role of ch-MYC-AS1 in antiviral defense against ALV-J infection and regulation of c-Myc protein, we performed pull-down assays with biotinylated ch-MYC-AS1 lncRNA in combination with mass spectrometry (MS) to search for potential ch-MYC-AS1-interacting proteins. MS analysis revealed that ch-MYC-AS1 may interact with Annexin A2 (ANXA2), which is a membrane-associated protein with a wide range of intracellular functions and a recurrent host factor in a variety of viral infection (25). As a novel ALV-J receptor, ANXA2 can permit the entry of ALV-J into its non-permissible cells (20). Thus, we focused on the interaction between ch-MYC-AS1 and ANXA2 in the following study.

RNA pull-down assay and western blot analysis validated that ANXA2 binds specifically to ch-MYC-AS1 RNA (Fig. 5A). Results from RIP assays showed that ch-MYC-AS1 was enriched with anti-ANXA2 antibody, which further confirmed the interaction between ch-MYC-AS1 and ANXA2 (Fig. 5B). Moreover, RNA FISH combined with confocal immunofluorescence showed that ch-MYC-AS1 RNA co-localized with ANXA2 in the cytoplasm and the nucleus of HD11 cells (Fig. 5C). Taken together, the data indicate that ch-MYC-AS1 directly binds to ANXA2.

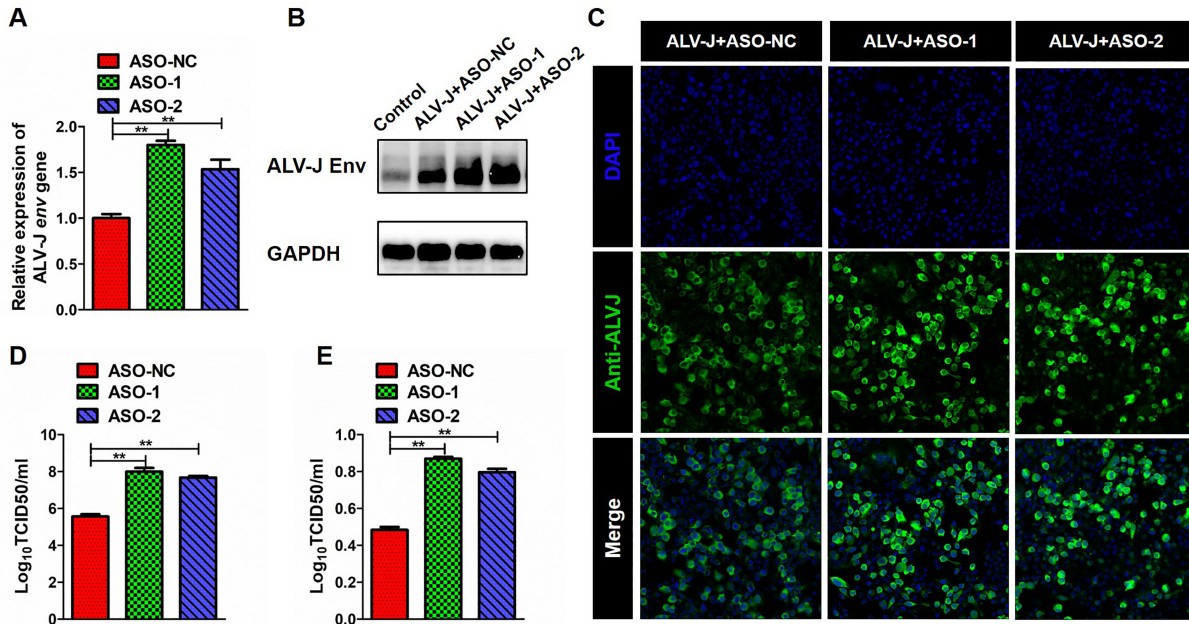

**FIG 3** Knockdown of ch-MYC-AS1 promoted the proliferation of ALV-J in chicken macrophages. HD11 cells were first transfected with the control or ch-MYC-AS1 antisense oligonucleotide (ASO) for 12 h and then infected with the ALV-J virus at an MOI of 5 for another 48 h. RT-qPCR (A), western blotting (B), and TCID$_{50}$ (C) analysis of ALV-J *env* gene expression in HD11 cells. (D) ELISA analysis of ALV-J p27 protein expression in the supernatant of HD11 cells. (E) Confocal immunofluorescence microscopy analysis of ALV-J *env* gene expression in HD11 cells. ASO refers to different antisense oligonucleotide sequences targeting ch-MYC-AS1, used to knock down ch-MYC-AS1 expression. Error bars represent the SEM, $n = 3$. *$P < 0.05$ and **$P < 0.01$ (two-tailed Student's *t*-test).

## Annexin A2 can interact with c-Myc protein

As a novel RNA-binding protein, ANXA2 not only can bind directly to proto-oncogene c-myc mRNA and up-regulates c-Myc protein (21) but also interacts with c-Myc protein and inhibits ubiquitin-dependent proteasomal degradation of c-Myc protein in the nucleus (26). To further elucidate the functional interplay between Annexin A2 and c-Myc, we investigated whether these two proteins physically interact in chicken macrophages.

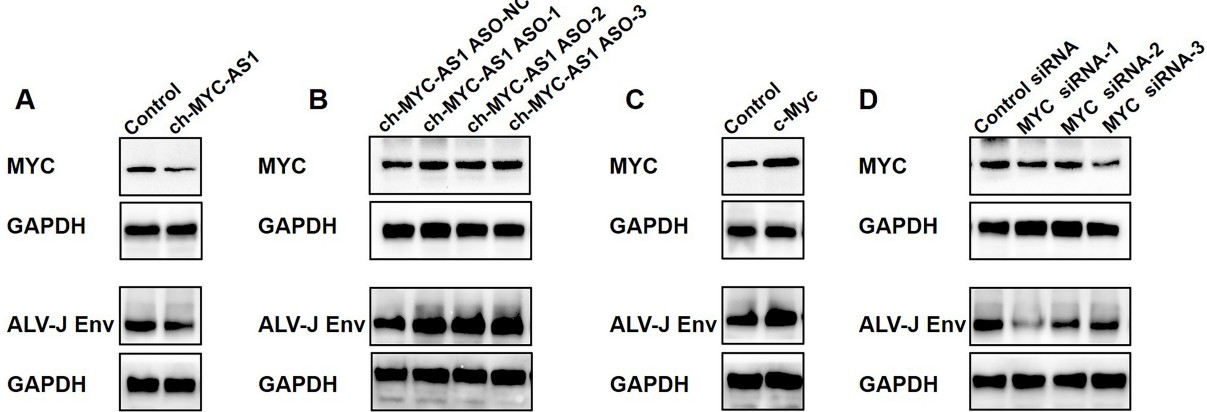

**FIG 4** ch-MYC-AS1 modulates ALV-J proliferation through c-Myc protein. (A) Western blotting analysis of MYC and ALV-J *env* gene expression in HD11 cells was first infected with the ALV-J virus at an MOI of 5 for 12 h and then transfected with pcDNA3.1-EGFP or pcDNA3.1-ch-MYC-AS1 plasmid for another 36 h. (B) Western blotting analysis of MYC and ALV-J *env* gene expression in HD11 cells was first transfected with the control or ch-MYC-AS1 ASO for 12 h and then infected with the ALV-J virus at an MOI of 5 for another 48 h. (C) Western blotting analysis of MYC and ALV-J *env* gene expression in HD11 cells was first infected with the ALV-J virus at an MOI of 5 for 12 h and then transfected with pcDNA3.1-EGFP or pcDNA3.1-ch-MYC plasmid for another 36 h. (D) Western blotting analysis of MYC and ALV-J *env* gene expression in HD11 cells was first transfected with the control or MYC siRNA for 12 h and then infected with the ALV-J virus at an MOI of 5 for another 48 h.

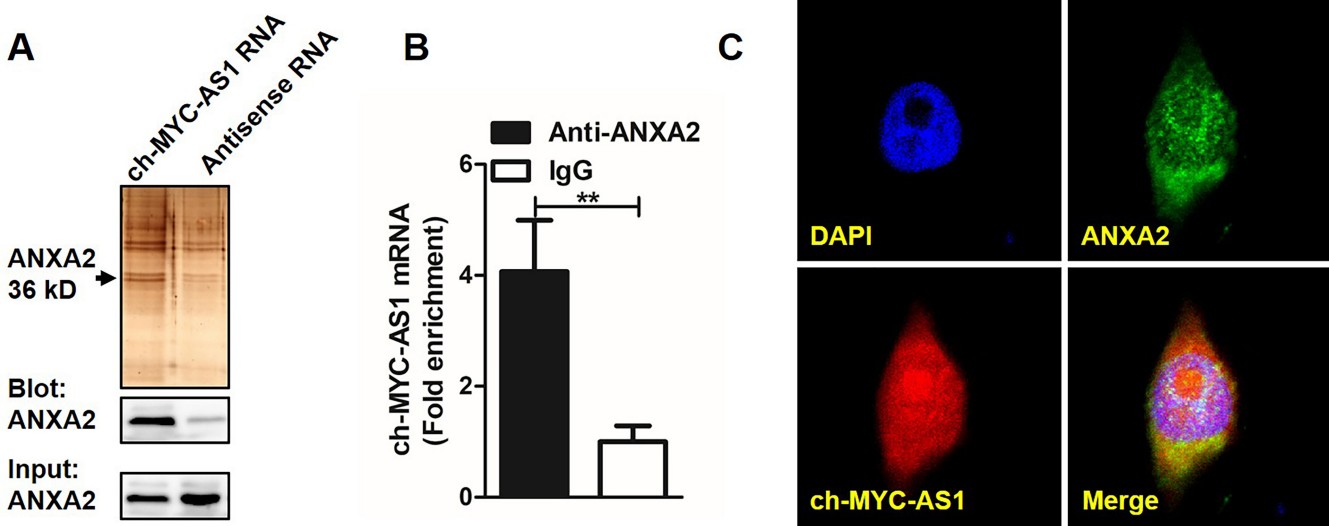

FIG 5 The lncRNA ch-MYC-AS1 directly interacts with ANXA2 protein in HD11 cells. (A) RNA pull-down assay using lysates from HD11 cells. Specific bands were excised and identified by mass spectrometry (MS; upper panel), or probed via immunoblotting for ANXA2 (lower panel). (B) RT–qPCR analysis of ch-MYC-AS1 enriched by ANXA2-specific antibody compared to immunoglobulin G (IgG) control in RIP assays using HD11 cell lysates. (C) Subcellular colocalization of ch-MYC-AS1 and ANXA2 in HD11 cells visualized by a combined RNA FISH assay for ch-MYC-AS1 and immunofluorescence staining for ANXA2, imaged using confocal microscopy.

Co-immunoprecipitation assays performed in HD11 cell lysates revealed a robust endogenous interaction between ANXA2 and c-Myc. Immunoprecipitation of c-Myc efficiently pulled down ANXA2 (Fig. 6A), and reciprocally, ANXA2 immunoprecipitation enriched c-Myc protein (Fig. 6B), confirming a specific ANXA2-c-Myc complex. We further visualized the subcellular distribution of the ANXA2–c-Myc complex using confocal immunofluorescence microscopy. In control HD11 cells, both proteins showed diffuse localization throughout the cytoplasm and nucleus, with areas of partial overlap (Fig. 6C). Notably, overexpression of ch-MYC-AS1 markedly reduced the nuclear accumulation of ANXA2 and c-Myc, resulting in their predominant retention in the cytoplasm (Fig. 6D). Together, these data confirm that ANXA2 specifically binds c-Myc protein in avian macrophages and suggest that ch-MYC-AS1 expression modulates the subcellular localization of this complex.

## ch-MYC-AS1 inhibits ALV-J replication by impairing nuclear translocation of the ANXA2–c-Myc complex

Having established that ch-MYC-AS1 modulates c-Myc activity by binding ANXA2, we next investigated whether this interaction influences the subcellular localization of the ANXA2–c-Myc complex. RNA FISH combined with confocal immunofluorescence revealed that ch-MYC-AS1 overexpression significantly reduced the nuclear accumulation of ANXA2 in HD11 cells (Fig. 7A). Conversely, ALV-J infection promoted ANXA2 nuclear translocation (Fig. 7B). Consistent with these observations, western blot analysis of subcellular fractions confirmed that ch-MYC-AS1 overexpression attenuated the infection-induced nuclear localization of ANXA2 (Fig. 7B). Based on these findings, we hypothesized that the ANXA2–c-Myc axis plays a functional role in ALV-J infection. Indeed, overexpression of ANXA2 abolished the suppressive effect of ch-MYC-AS1 on ALV-J replication in both HD11 and DF-1 cells (Fig. 7C and D), paralleling the results observed with c-Myc overexpression. Collectively, these data indicate that ch-MYC-AS1 binding to ANXA2 disrupts the nuclear import of the ANXA2–c-Myc complex, thereby impairing the pro-viral functions of c-Myc during ALV-J infection.

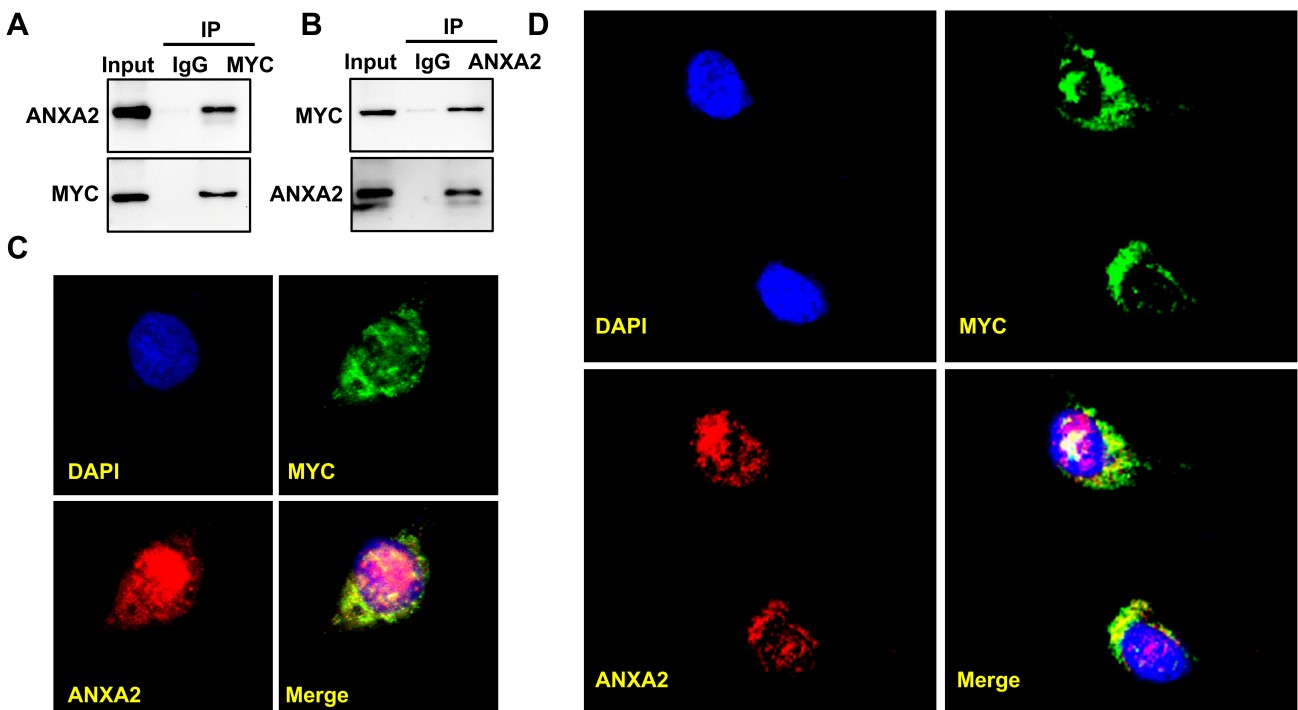

**FIG 6** ANXA2 directly interacts with MYC protein in HD11 cells. (A) Co-immunoprecipitation (Co-IP) of endogenous proteins from HD11 cell lysates using an antibody against c-Myc or control IgG, followed by immunoblotting for ANXA2. (B) Reciprocal Co-IP using an antibody against ANXA2 or control IgG, followed by immunoblotting for c-Myc. (C) Confocal immunofluorescence microscopy images of HD11 cells showing subcellular localization of endogenous ANXA2 and c-Myc. (D) Confocal immunofluorescence microscopy images of ch-MYC-AS1-highly expressed HD11 cells showing subcellular localization of endogenous ANXA2 and c-Myc.

## DISCUSSION

Retroviruses have garnered significant attention due to their unique replication cycle, oncogenic potential, and role in AIDS. Oncogenic retroviruses, in particular, were instrumental in identifying c-myc as a pivotal oncogene (27). Recent research demonstrates that c-Myc initiates and sustains tumorigenesis partly by globally dysregulating antitumor immunity—specifically through modulating the innate immune checkpoint CD47 and the adaptive immune checkpoint PD-L1 (8, 28). c-Myc overexpression enables tumors to evade immune destruction, a core hallmark of cancer (29), positioning c-Myc inhibition as a rational therapeutic strategy for novel anticancer agents (30).

The growing understanding of epigenetic processes in immune cell function and antitumor immunity has spurred interest in combining epigenetic therapy with immunotherapy (9). Notably, combined epigenetic approaches depleting c-Myc can reverse tumor immune evasion (16), suggesting unexplored epigenetic regulation of c-Myc. This study reveals that epigenetic silencing of the c-myc proto-oncogene is associated with its derived natural antisense RNA, ch-MYC-AS1. We identify ch-MYC-AS1 as a DNA methylation-regulated antisense noncoding RNA that significantly suppresses c-Myc expression and exhibits intrinsic antiviral activity against oncogenic retroviruses. Mechanistically, ch-MYC-AS1 disrupts retroviral biosynthesis primarily by interfering with MYC-modulated glycolysis.

We employed avian leukosis virus (ALV), a historically pivotal model in viral oncology (31), to dissect these mechanisms. ALV research underpinned three Nobel Prize-winning discoveries: virus-induced avian tumors (1966), reverse transcriptase (1975), and the Src oncogene (1989). The c-myc proto-oncogene was first identified as a common ALV integration site and driver of ALV-induced lymphomagenesis (3, 4). ALV integration

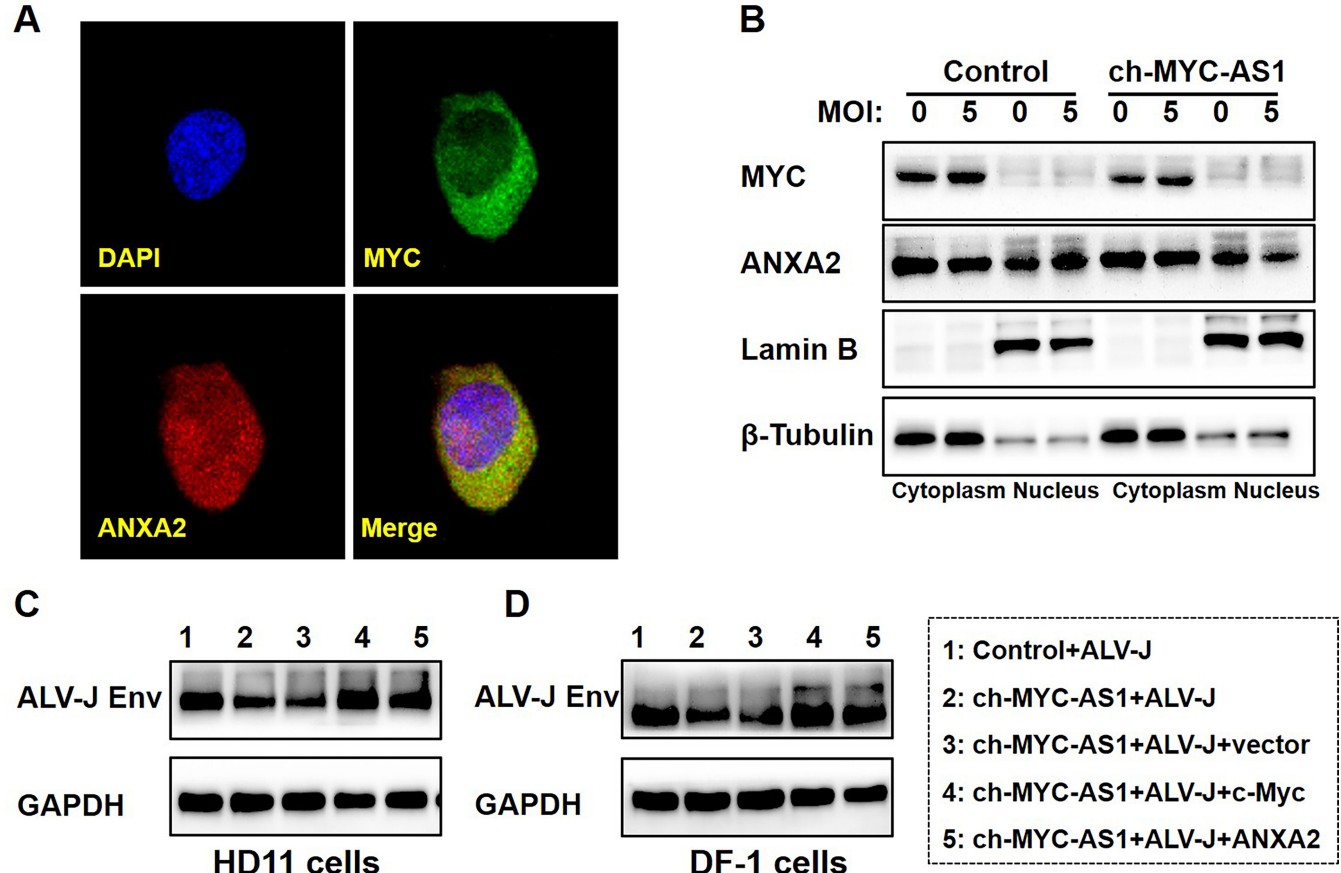

**FIG 7** ch-MYC-AS1 impairs ALV-J replication by inhibiting the nuclear translocation of the ANXA2–c-Myc complex. (A) Subcellular colocalization of ch-MYC-AS1 and ANXA2 in ch-MYC-AS1-highly expressed HD11 cells visualized by a combined RNA FISH assay for ch-MYC-AS1 and immunofluorescence staining for ANXA2, imaged using confocal microscopy. (B) Western blot analysis of MYC, ANXA2, beta Tubulin, and Lamin B1 protein levels in cytoplasmic and nuclear fractions of HD11 cells, which were infected with the ALV-J virus at an MOI of 0 or 5 for another 12 h and then transfected with pcDNA3.1-EGFP or pcDNA3.1-ch-MYC-AS1 plasmid for another 36 h. Western blotting analysis of ALV-J *env* gene expression in ch-MYC-AS1-highly expressed HD11 (C) or DF-1 (D) cells was first infected with the ALV-J virus at an MOI of 5 for 12 h and then transfected with the control or pcDNA3.1-ch-MYC or pcDNA3.1-ANXA2 plasmid for another 36 h.

deregulates c-myc, altering gene expression in the bursa of Fabricius and triggering bursal-derived B-cell lymphoma in susceptible chickens (32–34). Resistant chickens, however, mitigate ALV-driven c-Myc overexpression despite similar viral loads and integration frequencies (35), permitting normal B-cell differentiation (35, 36). We demonstrate that ch-MYC-AS1 acts as a natural barrier suppressing c-Myc. Critically, ALV-J proliferation was inhibited by c-Myc knockdown (via ch-MYC-AS1 overexpression or RNAi) but enhanced by c-Myc overexpression in chicken cells. Given that c-myc is a key ALV-J integration site (3), and that its dysregulation is central to ALV-J pathogenesis (37), ch-MYC-AS1-mediated c-Myc inhibition represents a potent antiviral strategy.

A key finding is that ch-MYC-AS1 impairs c-Myc protein expression through at least two mechanisms. First, ch-MYC-AS1 directly suppresses c-Myc transcription, potentially via RNA interference or epigenetic mechanisms. This is supported by the activation of ch-MYC-AS1 and concurrent suppression of c-Myc protein following treatment with demethylating agents. Second, ch-MYC-AS1 interferes with the nuclear function of ANXA2, an RNA-binding protein that stabilizes c-Myc mRNA and elevates c-Myc protein levels (21). This role of ANXA2 depends on its nuclear translocation, which requires Src-kinase-mediated phosphorylation at Tyr23 (26). In the nucleus, phosphorylated ANXA2 (Tyr23) interacts with MYC and inhibits its ubiquitin-dependent proteasomal degradation (26). ch-MYC-AS1 likely disrupts this pathway by impeding the phosphory-

lation or nuclear entry of ANXA2, thereby reducing nuclear ANXA2 and c-Myc protein levels.

ch-MYC-AS1 may exert additional antiviral effects by binding ANXA2, which has been identified as a functional receptor essential for ALV-J viral entry (20). Previous studies demonstrated that ANXA2 overexpression promoted, while its knockdown inhibited, ALV-J proliferation in both DF-1 cells and chicken macrophages (20). It is therefore plausible that ch-MYC-AS1 binding to ANXA2 could interfere with ALV-J entry into host cells, thereby contributing to its overall antiviral activity. Collectively, our results support a model in which ALV-J infection induces ANXA2 nuclear translocation and activates c-Myc-mediated signaling, whereas ch-MYC-AS1 counteracts this process by disrupting the ANXA2–c-Myc axis and impairing viral biosynthesis.

Our findings suggested that ch-MYC-AS1 may be developed into RNA therapeutics to inhibit ALV-J replication in poultry. Furthermore, the conserved role of the MYC–ANXA2 axis in viral infections highlights its potential as a target for broad-spectrum antiviral strategies. While our findings indicate that ch-MYC-AS1 binding hinders the nuclear accumulation of ANXA2, the precise mechanism underlying this blockade remains to be elucidated. ch-MYC-AS1 could potentially mask the nuclear localization signals of ANXA2, disrupt its phosphorylation by kinases such as Src, or competitively inhibit its interaction with nuclear import adaptors like importins. Further investigation using phospho-specific antibodies, co-immunoprecipitation of nuclear transport components, and site-directed mutagenesis of ANXA2 will be essential to clarify this mechanism.

## Conclusion

This study reveals a novel antiviral mechanism mediated by the c-Myc-derived antisense lncRNA ch-MYC-AS1, which effectively suppresses the proliferation of the oncogenic retrovirus ALV-J through disruption of the ANXA2–c-Myc functional axis. These findings enhance our understanding of both retroviral pathogenesis and the regulatory roles of antisense noncoding RNAs, offering valuable insights for the development of targeted therapies against retroviral infections and MYC-driven malignancies.

## ACKNOWLEDGMENTS

This work was supported by grants from the National Natural Science Foundation of China grant (32272860), the Earmarked Fund for Modern Agro-industry Technology Research System in Waterfowl Industry (NO. CARS-42-3), and Priority Academic Program Development of Jiangsu Higher Education Institutions (Animal Science).

S.F.: writing—review and editing, writing—original draft, methodology, investigation, and formal analysis. W.Z.: writing—review and editing, writing—original draft, methodology, investigation, and formal analysis. X.H.: methodology, investigation, and formal analysis. Y.G.: methodology, investigation, and formal analysis. G.P.: methodology, investigation, and formal analysis. Y.Z.: writing—review and editing. W.Z.: funding acquisition and writing—review and editing. G.C.: funding acquisition and writing—review and editing. Q.X: funding acquisition and writing—review and editing.

## AUTHOR AFFILIATIONS

[1]Jiangsu Key Laboratory for Animal Genetic, Breeding and Molecular Design, College of Animal Science and Technology, Yangzhou University, Yangzhou, Jiangsu, China
[2]Institute of Epigenetics and Epigenomics, College of Animal Science and Technology, Yangzhou University, Yangzhou, Jiangsu, China
[3]School of Project Management, Faculty of Engineering, The University of Sydney, Sydney, New South Wales, Australia
[4]Joint International Research Laboratory of Agricultural and Agri-Product Safety, Ministry of Education of China, Yangzhou University, Yangzhou, Jiangsu, China

## AUTHOR ORCIDs

Xuming Hu ⓘ http://orcid.org/0000-0002-6973-8252

Qi Xu ⓘ http://orcid.org/0000-0003-2791-0429

## FUNDING

| Funder | Grant(s) | Author(s) |
|---|---|---|
| National Natural Science Foundation of China | 32272860 | Qi Xu |
| Earmarked Fund for Modern Agro-industry Technology Research System | NO. CARS-42-3 | Qi Xu |

## AUTHOR CONTRIBUTIONS

Suyu Fan, Formal analysis, Investigation, Methodology, Writing – original draft, Writing – review and editing | Weiyi Zhou, Formal analysis, Investigation, Methodology, Writing – original draft, Writing – review and editing | Xuming Hu, Formal analysis, Investigation, Methodology | Yingjie Gu, Formal analysis, Investigation, Methodology | Guangzhong Peng, Formal analysis, Investigation, Methodology | Yu Zhang, Writing – review and editing | Wenming Zhao, Funding acquisition, Writing – review and editing | Guohong Chen, Funding acquisition, Writing – review and editing | Qi Xu, Funding acquisition, Writing – review and editing

## DATA AVAILABILITY

All relevant data are within the paper.

## ADDITIONAL FILES

The following material is available online.

Open Peer Review

**PEER REVIEW HISTORY (review-history.pdf).** An accounting of the reviewer comments and feedback.

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
