## [Reviewer comments · Microbiology Spectrum]

Microbiology Spectrum

LncRNA ch-MYC-AS1 restricts ALV-J replication by disrupting the ANXA2-c-Myc oncogenic axis

Suyu Fan, Xuming Hu, Yingjie Gu, Guangzhong Peng, Yu Zhang, Wenming Zhao, Guohong Chen, and Qi Xu

Corresponding Author(s): Qi Xu, Yangzhou University

Review Timeline:

Submission Date:	December 2, 2025
Editorial Decision:	January 3, 2026
Revision Received:	January 20, 2026
Accepted:	February 1, 2026

Editor: Bin Zhu

Reviewer(s): Disclosure of reviewer identity is with reference to reviewer comments included in decision letter(s). The following individuals involved in review of your submission have agreed to reveal their identity: Wentao Li (Reviewer #2)

Transaction Report:

DOI: <https://doi.org/10.1128/spectrum.03880-25>

Re: Spectrum03880-25 (**LncRNA ch-MYC-AS1 restrains ALV-J replication through interferes with ANXA2-c-Myc oncogenic axis**)

Dear Prof. Qi Xu:

Thank you for the privilege of reviewing your work. Below you will find my comments, instructions from the Spectrum editorial office, and the reviewer comments.

Revision Guidelines

Sincerely,
Bin Zhu
Editor
Microbiology Spectrum

Reviewer #1 (Comments for the Author):

This is a well-executed study that provides significant insights into antiviral innate immunity. The authors identify a novel lncRNA, ch-MYC-AS1, and systematically demonstrate its role in restricting ALV-J replication via disruption of the ANXA2-c-Myc axis. The experimental design is rigorous, incorporating multiple complementary approaches, including RACE, RNA pull-down, RIP, and subcellular localization assays, to convincingly validate the molecular mechanism. The findings not only reveal a new

layer of epigenetic regulation in retroviral defense but also offer promising therapeutic implications for targeting MYC-driven pathologies. Well, several concerns should be addressed before publication.

1. The title clearly reflects the study's findings. However, for enhanced impact and conciseness, consider shortening it slightly. For example: "LncRNA ch-MYC-AS1 Restricts ALV-J Replication by Disrupting the ANXA2-c-Myc Oncogenic Axis."
2. Please ensure consistency by using the keyword "ch-MYC-AS1" throughout, matching the gene nomenclature in the text.
3. To better highlight the study's innovation, explicitly state the specific knowledge gap before presenting the aim. For example: "However, whether antisense noncoding RNAs from the c-myc locus possess intrinsic antiviral functions against retroviruses that dysregulate this oncogene remains unexplored."
4. The mechanistic core of the study revolves around ANXA2. Currently, the Introduction does not introduce this protein. Introduce ANXA2 with a brief sentence noting its dual role as a cellular receptor for ALV-J entry and an RNA-binding protein regulating c-Myc expression. This will help readers immediately grasp the significance of the discovered axis.
5. In Methods sections 2.3 and 2.4, the specific time intervals (e.g., "for 12 h") between viral infection and transfection are stated. Adding a short rationale (e.g., "...to allow for viral adsorption or "...to ensure stable gene expression before subsequent manipulation) would enhance the methodological clarity and reproducibility.
6. When cautiously discussing ch-MYC-AS1 potentially interfering with ANXA2 phosphorylation, consider briefly citing relevant literature on Src-mediated phosphorylation of ANXA2.
7. In Figure 6A, the Co-IP results are incomplete. The input group lacks a housekeeping protein control and does not show endogenous expression levels of target proteins; therefore, these data should be supplemented.

Reviewer #2 (Comments for the Author):

In this manuscript, Fan et al. identified and characterized a novel antisense long noncoding RNA (lncRNA), ch-MYC-AS1, which is transcribed from the chicken c-myc proto-oncogene locus. The authors demonstrated that ch-MYC-AS1 suppresses avian leukosis virus subgroup J (ALV-J) replication via a specific molecular mechanism: it binds to annexin A2 (ANXA2), inhibits ANXA2 nuclear translocation, and thereby disrupts the oncogenic ANXA2-c-Myc axis. Collectively, the studies provide convincing evidence that this novel transcript is present in chickens and can modulate ALV-J replication. My comments and critiques:

1. The mechanism through which ch-MYC-AS1 inhibits ANXA2 nuclear translocation warrants further investigation. Future studies could examine whether ch-MYC-AS1 affects ANXA2 post-translational modifications (e.g., phosphorylation) or its interaction with nuclear import proteins. Clarifying these points would help refine the proposed regulatory pathway.
2. The manuscript would benefit from a thorough language edit to enhance clarity and stylistic precision, particularly by varying repetitive phrasing (e.g., "further confirmed"). Please also ensure consistent use of the term "ch-MYC-AS1" throughout the text.
3. The introduction provides a solid background on c-Myc and retroviruses. To better contextualize the study's novelty, it would be valuable to briefly summarize the current understanding about lncRNA functions in avian antiviral immunity.
4. In the RNA-FISH and co-localization experiments, have you considered including a negative control (e.g., scrambled probe) to further validate the specificity of the ch-MYC-AS1 signal?
5. The discussion provides a thoughtful interpretation of the data within the existing literature. To enhance the impact, the authors could further explore the translational potential of ch-MYC-AS1—for instance, its applicability as a therapeutic target in poultry or as a broader antiviral strategy.
6. The abstract effectively summarizes the key findings. However, it would be strengthened by briefly specifying the model system used, such as chicken macrophage HD11 cells.
7. The manuscript does not provide the nucleotide sequences for the "ch-MYC-AS1 antisense oligonucleotide (ASO)" (used in Fig. 3, 4B) or the "MYC siRNA" (used in Fig. 4D). The Methods section (2.4) only states that "Stealth RNAi[®] siRNAs specific for ch-MYC-AS1 or chicken MYC" were used, without any sequence details. Similarly, the term "ASO" is used without specification. This omission makes it impossible for other researchers to replicate the knockdown experiments, which are central to the study's conclusions.

This is a well-executed study that provides significant insights into antiviral innate immunity. The authors identify a novel lncRNA, ch-MYC-AS1, and systematically demonstrate its role in restricting ALV-J replication via disruption of the ANXA2-c-Myc axis. The experimental design is rigorous, incorporating multiple complementary approaches, including RACE, RNA pull-down, RIP, and subcellular localization assays, to convincingly validate the molecular mechanism. The findings not only reveal a new layer of epigenetic regulation in retroviral defense but also offer promising therapeutic implications for targeting MYC-driven pathologies. Well, several concerns should be addressed before publication.

1. The title clearly reflects the study's findings. However, for enhanced impact and conciseness, consider shortening it slightly. For example: "LncRNA ch-MYC-AS1 Restricts ALV-J Replication by Disrupting the ANXA2-c-Myc Oncogenic Axis."
2. Please ensure consistency by using the keyword "ch-MYC-AS1" throughout, matching the gene nomenclature in the text.
3. To better highlight the study's innovation, explicitly state the specific knowledge gap before presenting the aim. For example: "However, whether antisense noncoding RNAs from the c-myc locus possess intrinsic antiviral functions against retroviruses that dysregulate this oncogene remains unexplored."
4. The mechanistic core of the study revolves around ANXA2. Currently, the Introduction does not introduce this protein. Introduce ANXA2 with a brief sentence noting its dual role as a cellular receptor for ALV-J entry and an RNA-binding protein regulating c-Myc expression. This will help readers immediately grasp the significance of the discovered axis.
5. In Methods sections 2.3 and 2.4, the specific time intervals (e.g., "for 12 h") between viral infection and transfection are stated. Adding a short rationale (e.g., "...to allow for viral adsorption or "...to ensure stable gene expression before subsequent manipulation) would enhance the methodological clarity and reproducibility.
6. When cautiously discussing ch-MYC-AS1 potentially interfering with ANXA2 phosphorylation, consider briefly citing relevant literature on Src-mediated phosphorylation of ANXA2.

7. In Figure 6A, the Co-IP results are incomplete. The input group lacks a housekeeping protein control and does not show endogenous expression levels of target proteins; therefore, these data should be supplemented.

Dear Editor ,

We must express our appreciation to you and the reviewers for the valuable comments provided. We have carefully revised the manuscript according to the comments and provided point-by-point responses to all the questions raised by the reviewers below. I hope that this revised version is acceptable for publication in your journal.

Response to Reviewer 1

Comments

This is a well-executed study that provides significant insights into antiviral innate immunity. The authors identify a novel lncRNA, ch-MYC-AS1, and systematically demonstrate its role in restricting ALV-J replication via disruption of the ANXA2-c-Myc axis. The experimental design is rigorous, incorporating multiple complementary approaches, including RACE, RNA pull-down, RIP, and subcellular localization assays, to convincingly validate the molecular mechanism. The findings not only reveal a new layer of epigenetic regulation in retroviral defense but also offer promising therapeutic implications for targeting MYC-driven pathologies. Well, several concerns should be addressed before publication.

Comments 1: *The title clearly reflects the study's findings. However, for enhanced impact and conciseness, consider shortening it slightly. For example: "LncRNA ch-MYC-AS1 Restricts ALV-J Replication by Disrupting the ANXA2-c-Myc Oncogenic Axis."*

Response 1: Following your advice, we have changed the title "LncRNA ch-MYC-AS1 restrains ALV-J replication through interferes with ANXA2-c-Myc oncogenic axis" to "LncRNA ch-MYC-AS1 restricts ALV-J replication by disrupting the ANXA2-c-Myc oncogenic axis" in the revised manuscript.

Comments 2: *Please ensure consistency by using the keyword "ch-MYC-AS1" throughout, matching the gene nomenclature in the text.*

Response 2: Thank you for pointing this out. We have carefully checked the entire manuscript and standardized the term to "ch-MYC-AS1" in all instances (e.g., Abstract, Results, Figures, and Methods).

Comments 3: *To better highlight the study's innovation, explicitly state the specific knowledge gap before presenting the aim. For example: "However, whether antisense noncoding RNAs from the c-myc locus possess intrinsic antiviral functions against retroviruses that dysregulate this*

oncogene remains unexplored."

Response 3: Following your advice, we have modified the second paragraph of the Introduction to include the specified knowledge gap in the revised manuscript.

In the Introduction section:

"However, whether antisense noncoding RNAs from the c-myc locus possess intrinsic antiviral functions against retroviruses that dysregulate this oncogene remains unexplored.

In this study, we aimed to..."

Comments 4: *The mechanistic core of the study revolves around ANXA2. Currently, the Introduction does not introduce this protein. Introduce ANXA2 with a brief sentence noting its dual role as a cellular receptor for ALV-J entry and an RNA-binding protein regulating c-Myc expression. This will help readers immediately grasp the significance of the discovered axis.*

Response 4: Thanks for this valuable suggestion. We have added the following sentence to the Introduction in the revised manuscript.

In the Introduction section:

" Notably, Annexin A2 (ANXA2) serves as a key cellular receptor for ALV-J entry [20] and also functions as an RNA-binding protein implicated in post-transcriptional regulation, including that of c-Myc [21], highlighting its potential central role in virus-host interactions."

Comments 5: *In Methods sections 2.3 and 2.4, the specific time intervals (e.g., "for 12 h") between viral infection and transfection are stated. Adding a short rationale (e.g., "...to allow for viral adsorption or "...to ensure stable gene expression before subsequent manipulation) would enhance the methodological clarity and reproducibility.*

Response 5: Thanks for this comment and your concern. We have incorporated these key details into the Methods section in the revised manuscript.

In the Methods section:

Section 2.3: "... For viral infection experiments, HD11 cells were first infected with the ALV-J virus at a multiplicity of infection (MOI) of 0.2, 1 and 5 for 12 h to allow viral adsorption and entry."

Section 2.4: "... For viral infection experiments, HD11 cells were first transfected with ch-MYC-AS1 ASOs, chicken MYC siRNAs or the control using Lipofectamine™ RNAiMAX Transfection Reagent (Thermo Fisher Scientific, USA) for 12 h to allow for adequate expression prior to ALV-J infection."

Comments 6: *When cautiously discussing ch-MYC-AS1 potentially interfering with ANXA2 phosphorylation, consider briefly citing relevant literature on Src-mediated phosphorylation of ANXA2.*

Response 6: Following your advice, we have discussed this point in the Discussion section in the revised manuscript.

In the Discussion section:

"Second, ch-MYC-AS1 interferes with the nuclear function of ANXA2, an RNA-binding protein that stabilizes c-Myc mRNA and elevates c-Myc protein levels [21]. This role of ANXA2 depends on its nuclear translocation, which requires Src-kinase-mediated phosphorylation at Tyr23 [26]. In the nucleus, phosphorylated ANXA2 (Tyr23) interacts with MYC and inhibits its ubiquitin-dependent proteasomal degradation [26]. ch-MYC-AS1 likely disrupts this pathway by impeding the phosphorylation or nuclear entry of ANXA2, thereby reducing nuclear ANXA2 and c-Myc protein levels."

Comments 7: *In Figure 6A, the Co-IP results are incomplete. The input group lacks a housekeeping protein control and does not show endogenous expression levels of target proteins; therefore, these data should be supplemented.*

Response 7: Following your advice, we have supplemented the Co-IP data in Figure 6A.

Response to Reviewer 2

Comments

In this manuscript, Fan et al. identified and characterized a novel antisense long noncoding RNA (lncRNA), ch-MYC-AS1, which is transcribed from the chicken c-myc proto-oncogene locus. The authors demonstrated that ch-MYC-AS1 suppresses avian leukosis virus subgroup J (ALV-J) replication via a specific molecular mechanism: it binds to annexin A2 (ANXA2), inhibits ANXA2 nuclear translocation, and thereby disrupts the oncogenic ANXA2-c-Myc axis. Collectively, the studies provide convincing evidence that this novel transcript is present in chickens and can modulate ALV-J replication. My comments and critiques:

Comments 1: The mechanism through which ch-MYC-AS1 inhibits ANXA2 nuclear translocation warrants further investigation. Future studies could examine whether ch-MYC-AS1 affects ANXA2 post-translational modifications (e.g., phosphorylation) or its interaction with nuclear import proteins. Clarifying these points would help refine the proposed regulatory pathway.

Response 1: Thanks for this comment and your concern. We have expanded our speculation to incorporate this point in the Discussion section in the revised manuscript.

In the Discussion section:

"While our findings indicate that ch-MYC-AS1 binding hinders the nuclear accumulation of ANXA2, the precise mechanism underlying this blockade remains to be elucidated. ch-MYC-AS1 could potentially mask the nuclear localization signals of ANXA2, disrupt its phosphorylation by kinases such as Src, or competitively inhibit its interaction with nuclear import adaptors like importins. Further investigation using phospho-specific antibodies, co-immunoprecipitation of nuclear transport components, and site-directed mutagenesis of ANXA2 will be essential to clarify this mechanism."

Comments 2: The manuscript would benefit from a thorough language edit to enhance clarity and stylistic precision, particularly by varying repetitive phrasing (e.g., "further confirmed"). Please also ensure consistent use of the term "ch-MYC-AS1" throughout the text.

Response 2: Following your advice, we have corrected "diao" to "Diao" in the revised manuscript. The manuscript has undergone professional language editing to improve clarity, flow, and stylistic variation. Phrases like "further confirmed" have been replaced with alternatives such as "validated," "corroborated," or "provided additional evidence for." Furthermore, we have ensured the consistent use of "ch-MYC-AS1" throughout the text.

Comments 3: *The introduction provides a solid background on c-Myc and retroviruses. To better contextualize the study's novelty, it would be valuable to briefly summarize the current understanding about lncRNA functions in avian antiviral immunity.*

Response 3: Thank you for pointing this out. We have added a new sentence in the Introduction to address this point: " In chickens, several long non-coding RNAs (lncRNAs) serve as key regulators of antiviral immunity by modulating interferon responses and viral replication pathways [17-19]."

Comments 4: *In the RNA-FISH and co-localization experiments, have you considered including a negative control (e.g., scrambled probe) to further validate the specificity of the ch-MYC-AS1 signal?*

Response 4: Thank you for pointing this out. Indeed, in our preliminary validation, we performed the RNA-FISH and co-localization experiments using a scrambled negative control probe. As no specific fluorescence signal was detected under these control conditions, the data were not included in the final manuscript in order to maintain focus on the primary findings.

Comments 5: *The discussion provides a thoughtful interpretation of the data within the existing literature. To enhance the impact, the authors could further explore the translational potential of ch-MYC-AS1-for instance, its applicability as a therapeutic target in poultry or as a broader antiviral strategy.*

Response 5: Following your advice, we have expanded this point in the Discussion section in the revised manuscript.

In the Discussion section:

"Our findings suggested that ch-MYC-AS1 may be developed into RNA therapeutics to inhibit ALV-J replication in poultry. Furthermore, the conserved role of the MYC-ANXA2 axis in viral infections highlights its potential as a target for broad-spectrum antiviral strategies."

Comments 6: *The abstract effectively summarizes the key findings. However, it would be strengthened by briefly specifying the model system used, such as chicken macrophage HD11 cells.*

Response 6: Thanks for this comment and your concern. We have revised the abstract in the revised manuscript.

In the Abstract section:

" Here we show that ch-MYC-AS1, a novel lncRNA, restricts avian leukosis virus subgroup J (ALV-J) replication by targeting c-Myc protein expression using chicken macrophage HD11 cells."

Comments 7: *The manuscript does not provide the nucleotide sequences for the "ch-MYC-AS1 antisense oligonucleotide (ASO)" (used in Fig. 3, 4B) or the "MYC siRNA" (used in Fig. 4D). The Methods section (2.4) only states that "Stealth RNAi {trade mark, serif} siRNAs specific for ch-MYC-AS1 or chicken MYC" were used, without any sequence details. Similarly, the term "ASO" is used without specification. This omission makes it impossible for other researchers to replicate the knockdown experiments, which are central to the study's conclusions.*

Response 7: Thanks for this comment and your concern. We have provided the complete nucleotide sequences for ch-MYC-AS1 antisense oligonucleotide (ASO)" (used in Fig. 3, 4B) or the "MYC siRNA" (used in Fig. 4D) within the Methods in the revised manuscript.

In the Methods section:

"Gene-specific antisense oligonucleotides (ASOs) targeting ch-MYC-AS1 and small interfering RNAs (siRNAs) targeting chicken MYC were designed and synthesized by Guangzhou RiboBio Co., Ltd. (Guangzhou, China). The sequences were as follows:

ch-MYC-AS1 ASOs: ASO-1 (5'-GGACUCUGUGCUGGAUUCAG-3'),

ASO-2 (5'-CCUUCUCGUUGUUGGCCACC-3'),

ASO-3 (5'-UCCUCUGAGUCUAACGUGCG-3').

Chicken MYC siRNAs: siRNA-1 (5'-CCAGCAAGAACUACGAUUA-3'),

siRNA-2 (5'-CAGACUAAUCGCAGAGAAA-3'),

siRNA-3 (5'-AGAUCAGCAACAACCGAAA-3')."

In the Fig. 3 legend, we have also clarified that the "ASO" refers to different antisense oligonucleotide sequences targeting ch-MYC-AS1, used to knock down ch-MYC-AS1 expression.

Re: Spectrum03880-25R1 (LncRNA ch-MYC-AS1 restricts ALV-J replication by disrupting the ANXA2-c-Myc oncogenic axis)

Dear Prof. Qi Xu:

Your manuscript has been accepted, and I am forwarding it to the ASM production staff for publication. Your paper will first be checked to make sure all elements meet the technical requirements. ASM staff will contact you if anything needs to be revised before copyediting and production can begin. Otherwise, you will be notified when your proofs are ready to be viewed.

Sincerely,
Bin Zhu
Editor
Microbiology Spectrum

Reviewer #1 (Comments for the Author):

All of my concerns have been appropriately addressed.